# Dynamic Token Modulation and Expansion for Multi-Task Learning

## Abstract

Multi-Task Learning (MTL) aims to minimize negative transfer within a shared network. Common strategies involve separating task-generic and task-specific representations and coordinating them to work together effectively within MTL frameworks. However, the absence of a clear rule for determining task-specific network components challenges the design of efficient MTL architectures. Our method tackles negative transfer by employing token-based network expansion and modulation without directly modifying predefined architectures, making it adaptable to any transformer-based MTL architectures. To evaluate negative transfer, we treat tokens as parameters, assessing gradient conflicts during backpropagation. Conflicts between tasks are analyzed by examining the token's range space and null space. Based on conflict types, we expand the network following rules. If task-specific gradients clash in the tokens' range space, we modulate existing tokens to align their task gradients. Conversely, if the gradients conflict in the null space of tokens, we add new task-specific tokens, spanning a new feature space. Our approach effectively boosts multi-task performance across various datasets by being integrated into previous state-of-the-art multi-task architectures.

## 1 Introduction

Multi-task learning in computer vision is an essential technique for creating efficient and effective deep learning models that can work with a unified architecture for multiple tasks (Caruana, 1997), resulting in better generalization and faster convergence. Additionally, by combining related tasks into one model, the need for expensive computing and storage resources is reduced, making it a more viable option for a variety of applications.

MTL aims to minimize negative transfer (Crawshaw, 2020) across various tasks, as negative transfer occurs when learning one task hinders the performance of others. This can lead to a trade-off among tasks due to their distinct objectives. To address this, prior research on multi-task architectures predominantly concentrates on determining the type of information the architecture should learn for accurate predictions. Ye & Xu (2022b) classifies this information into three dimensions: task-generic representations, task-specific representations, and cross-task interactions. In previous studies (Eigen & Fergus, 2015; Xu et al., 2018; Vandenhende et al., 2020; Zhang et al., 2019; Dai et al., 2016; Ma et al., 2018; Simonyan & Zisserman, 2014; Zhang et al., 2014), a shared encoder is employed to learn generic representations, while task-specific features are refined in the decoder through cross-task interactions. Conversely, cross-talk architecture utilizes separate symmetrical networks for each task, incorporating cross-task interactions (Gao et al., 2019; Xu et al., 2018). Another approach (Maninis et al., 2019; Sun et al., 2021; Sinha et al., 2018; Fernando et al., 2017) involves dividing task-generic and task-specific information using task-specific modules.

Lately, multi-task architectures based on transformers have not only shown impressive performance across various tasks but have also excelled in the few-shot learning setting (Kim et al., 2023). These advancements draw inspiration from the success observed in the NLP domain (Shazeer et al., 2017). The two most prominent types of transformer-based multi-task paradigms are MoE (Riquelme et al., 2021; Zhang et al., 2022; Fan et al., 2022; Mustafa et al., 2022; Chen et al., 2023) and Task Prompter (Xu et al., 2023a;b; Ye & Xu, 2022b). MoE (Mixture of Experts) employs distinct specialized expert modules to learn various aspects of tasks. It utilizes a gating mechanism to decide which combination of experts should contribute to the final output. Task Prompter guides the model to learn

task-specific information by providing task-specific prompts. Previous research relies on manually designed modules, leading to a lack of generality in distinguishing shared and task-specific representations. MoE involves predefined number of expert modules into a backbone network. Similarly, Task Prompter requires predefined task prompts and modules that support their interaction.

In this context, a pivotal question arises: can a pre-defined module efficiently handle parameters, or is a pre-defined task space sufficient to encompass all task-specific information? From the analysis of previous works in transfer learning (Dwivedi & Roig, 2019), we identify three reasons why a predefined space cannot efficiently capture task-specific information. Firstly, the similarity between tasks changes as we move through the network's depth, implying that the extent of shared network elements should differ based on the network's depth. Secondly, the required task-specific space within the network for a task is not consistently uniform across depth. For tasks such as semantic segmentation, a substantial amount of space in deeper layers is required to leverage semantic information. In contrast, low-level vision tasks like surface normal estimation might necessitate more space in relatively shallower layers. Thirdly, these variations are dependent on the dataset being used. As a result, current multi-task architectures face inefficiencies due to the predefined modules for learning shared and task-specific information. In this study, we present a network expansion paradigm that can be applied to any transformer-based multi-task architecture to mitigate this inefficiency.

In order to improve the adaptability of a multi-task architecture by dynamically partitioning task-generic and task-specific representations, we focus on the concept of *conflicting gradients* (Yu et al., 2020). Conflicting gradients are recognized as a cause of negative transfer that emerges when the gradients of two tasks move in opposing directions. In contrast to previous approaches (Guangyuan et al., 2022) that transform shared parameters into task-specific ones by duplicating them, we leverage tokens of the transformer in our method. This choice not only enhances parameter efficiency but also extends applicability across various architectures. To expand the network based on tokens and prevent negative transfer by guaranteeing adequate space for tasks, we start by defining the token space as the output of each layer in the transformer block using singular value decomposition (SVD). Subsequently, we categorize gradient conflicts into two types: conflicts in the range space and null space of tokens. If task-specific gradients conflict within the token range space, we modulate tokens in that layer to align the gradients of different tasks. Conversely, if conflicts arise within the null space of tokens, we introduce new task-specific tokens to the network to learn new task-specific features. Importantly, our methods can be applied concurrently to previous multi-task architectures (Shazeer et al., 2017; Riquelme et al., 2021; Zhang et al., 2022; Fan et al., 2022; Mustafa et al., 2022; Chen et al., 2023; Xu et al., 2023a;b; Ye & Xu, 2022b) or network expansion methods (Guangyuan et al., 2022). In summary, our main contribution is three-fold:

- We introducea Dynamic Token Modulation and Expansion (DTME-MTL) approach for transformer-based multi-task architectures, which effectively reduces negative transfer caused by gradient conflicts. As far as we know, this is the first work dynamically expanding the network by manipulating tokens for MTL.

- We analyze conflicts between tasks in both token range space and null space, proposing diverse methodologies for token manipulation based on the nature of the conflict. If task-specific gradients conflict within the token range space, we modulate existing tokens in that layer. On the other hand, if conflicts arise within the null space, we introduce new task-specific tokens to the network. This approach, involving distinct response strategies for each conflict type, leads to the creation of an efficient network expansion system applicable to various existing multi-task architectures.

- DTME-MTL can be applied to existing state-of-the-art multi-task architectures in an off-the-shelf manner to enhance multi-task performance. We compare it with other off-the-shelf multi-task optimization methods to evaluate how effectively it mitigates negative transfer.

## 2 RELATED WORKS

**Multi-Task Learning in Vision Transformers.** Originally designed for NLP tasks, transformers have outperformed existing CNN models in various computer vision tasks. Attempts have been made to incorporate Vision Transformer (Dosovitskiy et al., 2020; Liu et al., 2021c; Wang et al., 2021a; Yang et al., 2021; Xie et al., 2021; Wang et al., 2021b) in MTL. MTFormer (Xu et al., 2022) employs a shared transformer encoder and decoder with a cross-task attention mechanism. MulT (Bhattacharjee et al., 2022) utilizes a shared attention mechanism to model task dependencies based

on the Swin transformer. InvPT (Ye & Xu, 2022a) focuses on global spatial position and multi-task context for dense prediction tasks through multi-scale feature aggregation. Mixture of Experts (MoE), inspired by the NLP domain, divides the model into predefined expert groups, adaptively shared or devoted to specific tasks during the learning phase (Riquelme et al., 2021; Zhang et al., 2022; Fan et al., 2022; Mustafa et al., 2022; Chen et al., 2023; Huang et al., 2024). Task prompter (Xu et al., 2023a;b; Ye & Xu, 2022b) uses task-specific tokens to encapsulate task-specific information and employs cross-task interactions to enhance multi-task performance. Prior studies need either a manually designed module to divide shared and task-specific representations, leading to a lack of generality. On the contrary, our methods can be applied to a diverse range of multi-task architectures, including those mentioned earlier.

**Multi-Task Optimization.** Optimizing the MTL aims to address negative transfer by adjusting the relative weighting of task losses or directly manipulating gradients. Task-dependent uncertainty (Kendall et al., 2018) is utilized to weigh the loss of multiple tasks. Liu et al. (2019) considers the rate of loss descent for achieving balance, while (Guo et al., 2018) prioritizes tasks based on difficulty. Recently, Liu et al. (2024) proposed updating task weights based on the loss history. In contrast, approaches like (Désidéri, 2012; Sener & Koltun, 2018; Yu et al., 2020; Liu et al., 2021a;b; Navon et al., 2022; Senushkin et al., 2023) directly modify task gradients to achieve the desired balance. PCGrad (Yu et al., 2020) analyzes negative transfer by identifying conflicting gradients in the shared parameters of the network. Recon (Guangyuan et al., 2022) transforms shared parameters directly into task-specific ones to handle conflicting gradients. Normalized gradients are employed to prevent spillover between tasks (Chen et al., 2018), whereas Chen et al. (2020) introduce stochasticity to the network's parameters based on the consistency in the sign of gradients. RotoGrad (Javaloy & Valera, 2021) rotates the feature space of the network to narrow the gap between tasks.

## 3 PRELIMINARIES

In multi-task learning, the network learns a set of tasks $\{\tau_i\}_{i=1}^{\mathcal{K}}$ jointly, where $\mathcal{K}$ is the number of tasks. Each task $\tau_i$ has its own loss function $\mathcal{L}_i$. The network parameter $\Theta$ can be classified into $\Theta = \{\Theta_s, \Theta_1, ..., \Theta_{\mathcal{K}}\}$ where $\Theta_s$ is shared parameter across all tasks and $\Theta_i$ is task-specific parameters devoted to task $\tau_i$. Then, the objective function of multi-task learning is to minimize the weighted sum of all tasks' losses: $\Theta^* = \arg\min_{\Theta} \sum_{i=1}^{\mathcal{K}} w_i \mathcal{L}_i(\Theta_s, \Theta_i)$ where $w_i$ represents the scale of the task-specific loss $\mathcal{L}_i$. Negative transfer between tasks occurs when the gradients of each objective point in different directions, a phenomenon called conflicting gradients (Yu et al., 2020).

**Definition 1** (Conflicting gradients). *Define $g_i$ as the gradient of task $\tau_i$ with respect to the shared parameters $\Theta_s$ as $g_i = \nabla_{\Theta_s} \mathcal{L}_i(\Theta_s, \Theta_i)$. Let $g_i$ and $g_j$ represent the gradients for a pair of tasks $\tau_i$ and $\tau_j$ where $i \neq j$. If $g_i \cdot g_j \leq 0$, these two gradients are termed conflicting gradients.*

The relationship between negative transfer and conflicting gradients is debated, with some studies taking opposing views. Jiang et al. (2024) presents a counterexample challenging the positive link between negative transfer and conflicting gradients in the context of auxiliary task learning. However, we adopt the conventional stance that conflicting gradients are widely seen as a key factor contributing to negative transfer in multi-task learning optimization (Désidéri, 2012; Sener & Koltun, 2018; Yu et al., 2020; Liu et al., 2021a;b; Navon et al., 2022; Senushkin et al., 2023; Jeong & Yoon, 2024), where tasks are primarily learned jointly rather than serving as auxiliary tasks for others. Existing methods that use pre-defined architectures for MTL have limitations in reducing negative transfer since they cannot preemptively prevent the occurrence of conflicting gradients. Guangyuan et al. (2022) involves transforming a shared layer into task-specific layers when conflicting gradients are detected in that layer. However, this method exhibits inefficiency in terms of the number of parameters, as it duplicates layers by a factor of the number of tasks, $\mathcal{K}$. In our approach, we adopt a more efficient token-based network expansion system instead of merely increasing the number of layers. Furthermore, we categorize gradient conflict into two types, presenting varied methodologies based on the nature of the conflict.

## 4 METHOD

As discussed in Section 3, conflicts can arise among gradients from task-specific losses, leading to negative transfer. In order to mitigate negative transfer by ensuring sufficient space for tasks, we

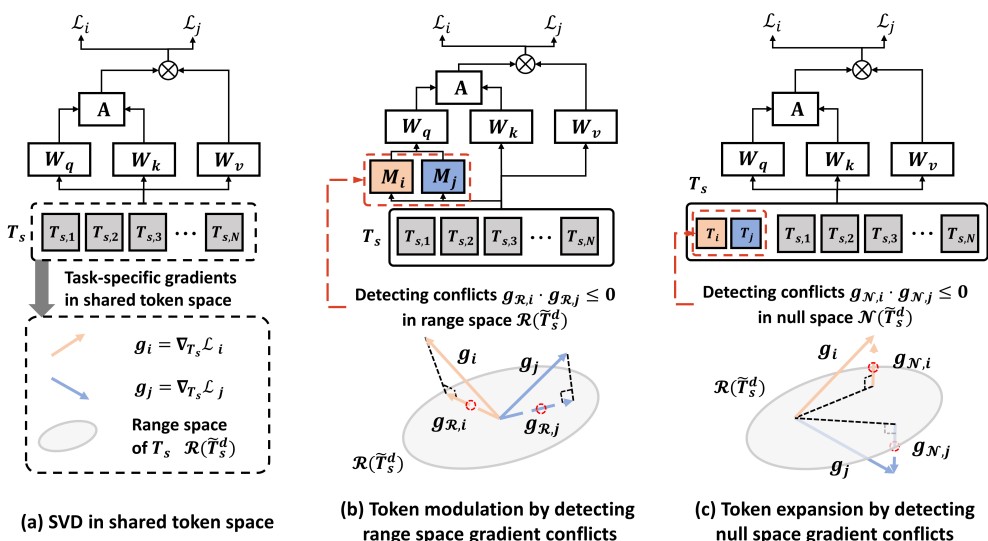

(a) SVD in shared token space

(b) Token modulation by detecting range space gradient conflicts

(c) Token expansion by detecting null space gradient conflicts

Figure 1: Framework overview of the proposed Dynamic Token Modulation and Expansion for MTL (DTME-MTL). (a) At each network layer, we compute the input token's range space $\mathcal{R}(\widetilde{\mathcal{T}}_s^d)$ and their task-specific gradients, determining principal vectors from the uncentered covariance of $\mathcal{T}_s$. (b) In cases where task-specific gradients conflict in the range space of $\widetilde{\mathcal{T}}_s^d$ (e.g. $g_{\mathcal{R},i} \cdot g_{\mathcal{R},j} \leq 0$), modulation is applied to $\mathcal{T}_s$ by introducing $\mathcal{M}_i$ and $\mathcal{M}_j$. (c) When task-specific gradients conflict within the null space of $\widetilde{\mathcal{T}}_s^d$ (e.g. $g_{\mathcal{N},i} \cdot g_{\mathcal{N},j} \leq 0$), task-specific tokens $\mathcal{T}_i$ and $\mathcal{T}_j$ are added.

adopt token-based network expansion. Initially, we define the token space as the output of each layer in the transformer block through singular value decomposition (SVD). Subsequently, we categorize conflicts in task-specific gradients into two types: conflicts in the range space of tokens and conflicts in the null space of tokens. Finally, based on the type of conflict, we introduce efficient token modulation and expansion techniques for transformer-based multi-task architectures.

## 4.1 DEFINING TOKEN SPACE USING SVD

In this section, we create a vector space consisting of shared tokens in a transformer, aiming to classify the types of conflicting gradients. More specifically, we approximate the range space and null space of the uncentered covariance of the tokens before applying our methods.

Let's consider a dataset $\{\mathcal{X}_l, \mathcal{Y}_l\}_{l=1}^n$, where $\mathcal{X}_l$ represents the input, $\mathcal{Y}_l$ denotes the label, and $n$ is the number of samples. Denote input shared token for a layer $d$ as $\mathcal{T}_s^{l,d} = \{\mathcal{T}_{s,1}^{l,d}, \mathcal{T}_{s,2}^{l,d}, ..., \mathcal{T}_{s,N}^{l,d}\}$ where $N$ is the total number of shared tokens in that layer. Every token $\mathcal{T}_s^{l,d} \in \mathbb{R}^p$ represents the output of the transformer layer $d-1$ for the corresponding input data $\mathcal{X}_l$, with $p$ denoting the size of $\mathcal{T}_s^{l,d}$. Let's consider a total of $D$ transformer layers. Next, the uncentered covariance of the token in layer $d$ (where $1 \leq d \leq D$) is as follows:

$$\widetilde{\mathcal{T}}_s^d = \frac{1}{n} \sum_{l=1}^n (\mathcal{T}_s^{l,d})(\mathcal{T}_s^{l,d})^T \tag{1}$$

To define the token space, we apply Singular Vector Decomposition to $\widetilde{\mathcal{T}}_s^d$. Following this, we can divide vector space formed by $\widetilde{\mathcal{T}}_s^d$ into its range space $\mathcal{R}(\widetilde{\mathcal{T}}_s)$ and null space $\mathcal{N}(\widetilde{\mathcal{T}}_s)$ depending on the magnitude of eigenvalue $\Lambda$. The process is illustrated below:

$$SVD(\widetilde{\mathcal{T}}_s^d) = \mathcal{U}, \Lambda, \mathcal{V} \quad \text{where} \quad \widetilde{\mathcal{T}}_s^d = \mathcal{U}\Lambda\mathcal{V}^T, \qquad \Lambda = \begin{bmatrix} \Lambda_\mathcal{R} & 0 \\ 0 & \Lambda_\mathcal{N} \end{bmatrix} \tag{2}$$

In this context, given that $\widetilde{\mathcal{T}}_s^d$ is a square matrix of dimensions $p \times p$, it implies that both $\mathcal{U}$ and $\mathcal{V}$ are square matrices as well, each with dimensions $p \times p$, and they are equal ($\mathcal{U} = \mathcal{V}$). Additionally, $\Lambda$ is a diagonal matrix.

$$SVD(\widetilde{\mathcal{T}_s^d}) = \mathcal{U}, \Lambda, \mathcal{V}$$

Figure 2: The process approximates the range and null spaces of $\tilde{\mathcal{T}}_s^d$ based on the proportion of total variance, $r$. In the SVD of $\tilde{\mathcal{T}}_s^d$, the matrix $\Lambda$ represents the diagonal matrix of eigenvalues. These eigenvalues are arranged in descending order, satisfying $\lambda_i \geq \lambda_j$ if $i < j$. If $r$ is greater than the sum up to $\lambda_m$ and smaller than the sum up to $\lambda_{m+1}$, then we select the set $\{\lambda_i\}_{i=1}^m$ as $\Lambda_{\mathcal{R}}$, and the remaining set $\{\lambda_i\}_{i=m+1}^p$ as $\Lambda_{\mathcal{N}}$.

From eq. (2), we obtain a mathematical tool to define the range and null space of the covariance of the token, $\widetilde{\mathcal{T}}_s^d$. To approximate the range space, we choose the eigenvalues $\Lambda_{\mathcal{R}}$ along with their corresponding eigenvectors from $\mathcal{U}_{\mathcal{R}}$. On the other hand, when approximating the null space, we should select the eigenvalues $\Lambda_{\mathcal{N}}$ and their corresponding eigenvectors from $\mathcal{U}_{\mathcal{N}}$. Ideally, we should choose eigenvalues that are exactly zero to form the null space. However, in practice, $\Lambda$ can not be precisely zero. Therefore, it's essential to establish a criterion for selecting the eigenvalue to distinguish between these two spaces. Instead of introducing a new manually designed rule for approximating each range and null space of $\widetilde{\mathcal{T}}_s^d$, we opt to directly employ the evaluation tool for the SVD process (Jollife & Cadima, 2016) as criteria for determining the range and null space of tokens. In assessing the accuracy of the SVD approximation, the proportion of total variance, denoted as $r$, has been employed as follows:

$$r = \frac{\sum_{\lambda \in diag\Lambda_{\mathcal{N}}} \lambda}{\sum_{\lambda \in diag\Lambda_{\mathcal{R}}} \lambda} \tag{3}$$

where each $\Lambda_{\mathcal{R}}$ and $\Lambda_{\mathcal{N}}$ represent submatrices of $\Lambda$ containing the eigenvalues of the range space and null space, respectively. The $diag$ function serves as an inverse matrix-to-vector operator, returning a vector containing the diagonal entries of the input matrix. In our approach, we employ eq. (3) to directly divide the range and null space of $\widetilde{\mathcal{T}}_s^d$. As depicted in Figure 2, the diagonal elements of the matrix $\Lambda$, obtained through the Singular Value Decomposition of $\tilde{\mathcal{T}}_s^d$, are arranged in descending order based on their magnitudes. We can select the index of the eigenvalue $m$ such that the sum of eigenvalues up to order $m$ is smaller than $r$, and the sum up to $m + 1$ is larger than $r$. This selected index serves as a boundary to divide the range space and null space of $\tilde{\mathcal{T}}_s^d$.

## 4.2 TYPES OF GRADIENT CONFLICTS

In Section 4.1, we create a $p$-dimensional vector space using the uncentered covariance of the shared token $\mathcal{T}_s$, linked to the input data set $\{\mathcal{X}\}_{l=1}^n$. This vector space is divided into the range and null space, with each space spanned by eigenvectors corresponding to singular values selected based on a specified ratio $r$. In the upcoming sections, we pinpoint the types of gradient conflict within the vector space we've constructed. We then address these conflicts adaptively by introducing token modulation and expansion techniques.

Using eq. (2) and eq. (3), we can partition the eigenvectors of the $p$-dimensional vector space into its range and null space, denoted as $\mathcal{U} = [\mathcal{U}_{\mathcal{R}}, \mathcal{U}_{\mathcal{N}}]$. Now, let's consider the shared tokens $T_s^{l,d}$ (where $l$ represents the input index and $d$ signifies the depth of the layer) as network parameters, for which we can compute gradients during the backpropagation process. For each task-specific loss $\mathcal{L}_i$, the task-specific gradient for $T_s^{l,d}$ is denoted as $g_i = \nabla_{\mathcal{T}_s^{l,d}} \mathcal{L}_i$. Consequently, we obtain task-specific gradients $\{g_i\}_{i=1}^{\mathcal{K}}$ corresponding to a set of losses $\{\mathcal{L}_i\}_{i=1}^{\mathcal{K}}$ for $T_s^{l,d}$ as illustrated in fig. 1-(a).

Each task-specific gradient $g_i$ can be decomposed into two components, $g_{\mathcal{R},i}$ and $g_{\mathcal{N},i}$, through projection onto the range and null space of $\tilde{\mathcal{T}}_s^d$, respectively. This breakdown is expressed as follows:

$$g_{\mathcal{R},i} = (\mathcal{U}_{\mathcal{R}}\mathcal{U}_{\mathcal{R}}^T)\nabla_{\mathcal{T}_s^{l,d}}\mathcal{L}_i \qquad\qquad g_{\mathcal{N},i} = (\mathcal{U}_{\mathcal{N}}\mathcal{U}_{\mathcal{N}}^T)\nabla_{\mathcal{T}_s^{l,d}}\mathcal{L}_i \tag{4}$$

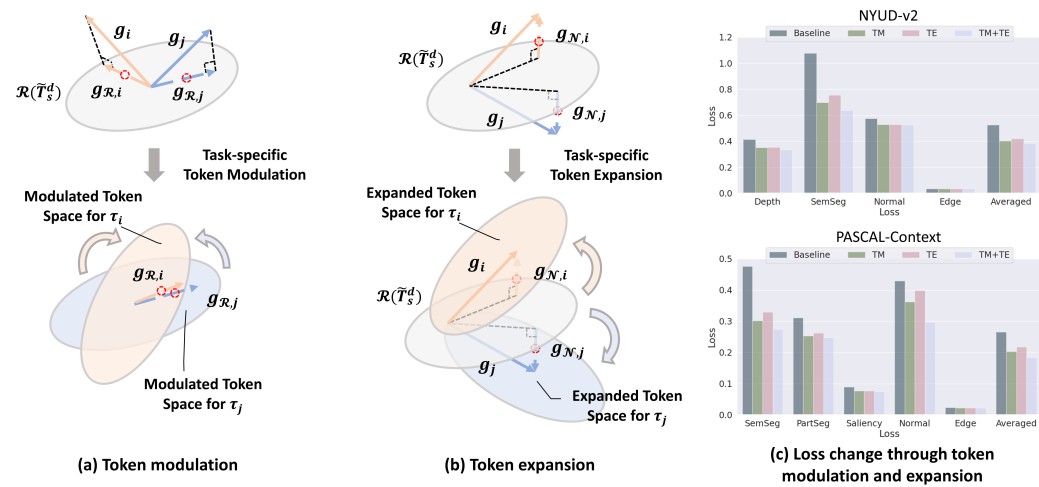

Figure 3: The figure illustrates the impact of token modulation and expansion on the token vector space. (a) Token modulation aligns task-specific gradients in the range space by adjusting the magnitude of shared tokens. (b) Token expansion broadens the range space by incorporating gradients in the null space, achieved through the addition of task-specific tokens. (c) Together, token modulation (TM) and expansion (TE) align task-specific loss to reduce multi-task loss.

$\mathcal{U}_\mathcal{R}$ and $\mathcal{U}_\mathcal{N}$ are orthogonal matrices that consist of eigenvectors of the range space and null space, respectively, with each column representing one eigenvector. Then, the matrices $(\mathcal{U}_\mathcal{R}\mathcal{U}_\mathcal{R}^T)$ and $(\mathcal{U}_\mathcal{N}\mathcal{U}_\mathcal{N}^T)$ function as projection operators onto the range and null spaces, respectively.

Building upon the concept of conflicting gradients outlined in Definition 1, we classify conflicts into two types based on the space in which they occur: range space conflicts and null space conflicts. Specifically, conflicts in the range space of tokens occur when $g_{\mathcal{R},i} \cdot g_{\mathcal{R},j} \leq 0$ for any pair of $i$ and $j$ where $i \neq j$. Likewise, conflicts in the null space of tokens emerge when $g_{\mathcal{N},i} \cdot g_{\mathcal{N},j} \leq 0$.

## 4.3 TOKEN MODULATION AND EXPANSION

From the types of gradient conflicts we defined in section 4.2, we present effective methods for token modulation and expansion in multi-task architectures based on transformers. For each transformer block with a depth of $d$, we can compute task-specific gradients $\{g_i\}_{i=1}^{\mathcal{K}}$ for the shared token $\mathcal{T}_s$. By utilizing eq. (4), we identify the specific types of conflicts that arise in a transformer block for a given input data $\mathcal{X}_i$. The extent of conflict is assessed by counting the occurrences of gradient conflicts across all data $\{\mathcal{X}_i, \mathcal{Y}_i\}_{i=1}^n$. To identify the layers with the most severe competition between tasks, we select the most conflicting layers to relieve negative transfer. The number of layers is a tunable hyperparameter controlled through network expansion.

**Token Modulation.** In situations where task-specific gradients conflict within the range space of $\widetilde{\mathcal{T}}_s^d$, such as $g_{\mathcal{R},i} \cdot g_{\mathcal{R},j} \leq 0$, modulators $\mathcal{M}_i$ and $\mathcal{M}_j$ are added after the shared token $\mathcal{T}_s^d$ as shown in fig. 1-(b). The token modulator $\mathcal{M}$ is a straightforward affine transformation that modulates the shared token $\mathcal{T}_s$ along the channel dimension. To elaborate, considering the embedding dimension of the transformer as $d_{model}$ (distinct from the layer depth $d$) and assuming the number of shared tokens is $N$, we can arrange $\mathcal{T}_s$ in the form $[\mathcal{T}_{s,1}, \ldots, \mathcal{T}_{s,N}]$. This arrangement turns $\mathcal{T}_s$ into a $d_{model} \times N$ matrix. The modulator $\mathcal{M}$ then performs the transformation $W[\mathcal{T}_{s,1}, \ldots, \mathcal{T}_{s,N}] + b$ using weight $W$ and bias $b$, both of which have dimensions $1 \times d_{model}$.

The intuition behind token modulation is to align task-specific gradients $\{g_i\}_{i=1}^{\mathcal{K}}$ by directly influencing the range space spanned by tokens, as shown in fig. 3-(a). During the learning process, the task-specific modulators $\{\mathcal{M}\}_{i=1}^{\mathcal{K}}$ learn to adjust this token space to align with task-specific gradients. This simple affine transformation is highly parameter-efficient in dealing with negative transfer resulting from conflicts in the range space of gradients.

**Token Expansion.** Similarly, in cases where task-specific gradients conflict within the null space of $\widetilde{\mathcal{T}}_s^d$, such as $g_{\mathcal{N},i} \cdot g_{\mathcal{N},j} \leq 0$, task-specific tokens $\mathcal{T}_i$ and $\mathcal{T}_j$ are added alongside shared tokens $\mathcal{T}_s$

---

**Algorithm 1:** Dynamic Token Modulation and Expansion for MTL

---

**Data:** Task $\{\tau_i\}_{i=1}^{\mathcal{K}}$, Loss function $\{\mathcal{L}_i\}_{i=1}^{\mathcal{K}}$, Dataset $\{\mathcal{X}_l, \mathcal{Y}_l\}_{l=1}^n$,
     Shared tokens $\mathcal{T}_s^{l,d} = \{\mathcal{T}_{s,i}^{l,d}\}_{i=1}^N$, Depth of the Network $D$

---

1 **for** *each layer of the network ($d \leftarrow 1$ to D)* **do**
2   Get tokens $\{\mathcal{T}_s^{l,d}\}_{l=1}^n$ for the layer $d$ corresponding to input $\{\mathcal{X}_l\}_{l=1}^n$
   $\widetilde{\mathcal{T}}_s^d = \frac{1}{n}\sum_{l=1}^n (\mathcal{T}_s^{l,d})(\mathcal{T}_s^{l,d})^T$     // Calculate uncentered covariance
3   $SVD(\widetilde{\mathcal{T}}_s^d) = \mathcal{U}, \Lambda, \mathcal{V}$       // Singular value decomposition
4   $\mathcal{U} = [\mathcal{U}_\mathcal{R}, \mathcal{U}_\mathcal{N}]$         // Divide range and null space
5   $\{g_{\mathcal{R},i}\}_{i=1}^{\mathcal{K}} = \{(\mathcal{U}_\mathcal{R}\mathcal{U}_\mathcal{R}^T)\nabla_{\mathcal{T}_s^{l,d}}\mathcal{L}_i\}_{i=1}^{\mathcal{K}}$    // Projection to range space
6   $\{g_{\mathcal{N},i}\}_{i=1}^{\mathcal{K}} = \{(\mathcal{U}_\mathcal{N}\mathcal{U}_\mathcal{N}^T)\nabla_{\mathcal{T}_s^{l,d}}\mathcal{L}_i\}_{i=1}^{\mathcal{K}}$    // Projection to null space
7   **if** $g_{\mathcal{R},i} \cdot g_{\mathcal{R},j} \leq 0$ **then**
8    └ Insert token modulators $\mathcal{M}_i$ and $\mathcal{M}_j$ prior to layer $d$
9   **if** $g_{\mathcal{N},i} \cdot g_{\mathcal{N},j} \leq 0$ **then**
10    └ Insert task-specific tokens $\mathcal{T}_i$ and $\mathcal{T}_j$ prior to layer $d$

---

as shown in fig. 1-(c). The task-specific tokens $\{\mathcal{T}_i\}_{i=1}^{\mathcal{K}}$ are concatenated with shared tokens before entering the transformer block. Consequently, each task-specific token acquires task-specific information within that layer. Specifically, in a standard transformer block, self-attention is performed for each pair of tokens in the form of $[\mathcal{T}_{s,1}, \ldots, \mathcal{T}_{s,N}] \times [\mathcal{T}_{s,1}, \ldots, \mathcal{T}_{s,N}]$. With token expansion, attention is extended to include $[\mathcal{T}_{s,1}, \ldots, \mathcal{T}_{s,N}] \times [\mathcal{T}_1, \ldots, \mathcal{T}_{\mathcal{K}}]$ on the output.

The rationale behind the token expansion is to widen the token space to incorporate task-specific gradients, as depicted in Figure 3-(b). Suppose we decompose task-specific gradients to extract the null space component of the token, and they indicate opposing directions, such as $g_{\mathcal{N},i} \cdot g_{\mathcal{N},j} \leq 0$. This suggests that the vector space spanned by the column vectors of $\mathcal{U}_\mathcal{R}$ cannot be updated to parameters where task-specific gradients point, as it exists outside of the token space. Expanding the token space by introducing task-specific tokens into the transformer layer, where conflicts in the null space arise, allows us to broaden the token spaces for different tasks. This enables each task-specific token space to include the task-specific gradients within the null space.

Token modulation and expansion work together to align the losses of various tasks, leading to improved multi-task performance, as shown in fig. 3-(c). While the proposed token modulation and expansion methods are intuitive, we also offer a theoretical analysis to support them. Theorem 1 demonstrates how applying token modulation to address gradient conflicts in the row space of $\tilde{\mathcal{T}}_s$ can reduce these conflicts and result in a lower multi-task loss.

**Theorem 1.** *Optimizing the token modulators $\{\mathcal{M}_i\}_{i=1}^{\mathcal{K}}$ reduces gradient conflicts in the row space of $\tilde{\mathcal{T}}_s$ and leads to a reduction in the multi-task loss.*

Similarly, in Theorem 2, we explain how expanding the token space to address gradient conflicts in the null space of $\tilde{\mathcal{T}}_s$ leads to a reduction in multi-task loss. All proofs can be found in Appendix A.

**Theorem 2.** *Token expansion using $\{\mathcal{T}_i\}_{i=1}^{\mathcal{K}}$ alleviates the increase in multi-task loss caused by gradient conflicts in the null space of $\tilde{\mathcal{T}}_s$.*

The complete procedure for the proposed DTME-MTL is outlined in Algorithm 1.

## 5 EXPERIMENTS

We conduct comprehensive experiments to show the effectiveness of the proposed Dynamic Token Modulation and Expansion for Multi-Task Learning (DTME-MTL).

**Datasets and Evaluation** Our method is evaluated on multi-task datasets: NYUD-v2 (Silberman et al., 2012), PASCAL-Context (Mottaghi et al., 2014) and Taskonomy (Zamir et al., 2018). Each of them with 4, 5, 11 tasks. To evaluate the performance of tasks, we employed widely used metrics. To evaluate the multi-task performance, we utilize the metric proposed by Maninis et al. (2019).

Table 1: We conduct an ablation study on dynamic token modulation and expansion, evaluating the multi-task performance of our method on NYUD-v2 and PASCAL-Context. The results of Token Extension (TE), Token Modulation (TM), and their combination (TE+TM) are presented. We employ a shared encoder and multiple decoders, using ViT-T (Dosovitskiy et al., 2020) as the backbone network. The gains are compared against single-task (ST) and multi-task (MT) scenarios.

| Model | NYUD-v2 | | | | PASCAL-Context | | | | |
|---|---|---|---|---|---|---|---|---|---|
| | Semseg mIoU ↑ | Depth RMSE ↓ | Normal mErr ↓ | Edge odsF ↑ | Semseg mIoU ↑ | Parsing mIoU ↑ | Saliency maxF ↑ | Normal mErr ↓ | Edge odsF ↑ |
| Baseline (ST) | 39.35 | 0.6611 | 22.14 | 59.68 | 67.96 | 58.90 | 83.76 | 15.65 | 47.70 |
| Baseline (MT) | 34.13 | 0.6732 | 22.51 | 55.30 | 54.47 | 51.48 | 82.04 | 16.22 | 41.28 |
| TM | 37.85 | 0.6490 | 21.75 | 56.92 | 64.28 | 55.10 | 83.02 | 15.40 | 45.80 |
| TE | 37.25 | 0.6553 | 21.87 | 57.00 | 60.51 | 54.00 | 82.85 | 15.55 | 44.98 |
| TM+TE | 38.27 | 0.6370 | 21.64 | 57.90 | 66.18 | 56.29 | 83.41 | 15.26 | 47.00 |
| Gain (vs. MT) | △4.14 | △0.0362 | △0.87 | △2.60 | △11.71 | △4.81 | △1.37 | △0.96 | △5.72 |
| $\triangle_m$ ↑ | 0.044 | | | | -1.289 | | | | |
| #Param ↑ (%) | 0.24 | | | | 0.30 | | | | |

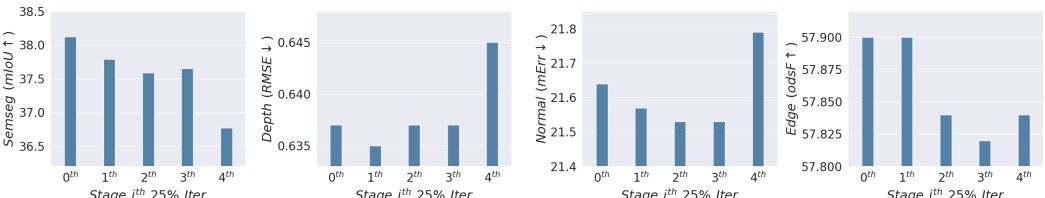

Figure 4: Task performance varies based on when we expand the network. To determine the optimal timing, we assess expansions at the beginning of training and at the end of each quarter iteration, monitoring the corresponding changes in performance.

It measures the per-task performance by averaging it with respect to the single-task baseline b, as shown in $\triangle_m = (1/T)\sum_{i=1}^{T}(-1)^{l_i}(M_{m,i} - M_{b,i})/M_{b,i}$ where $l_i = 1$ if a lower value of measure $M_i$ means better performance for task $i$, and 0 otherwise.

**Implementation Details.** For experiments, we adopt ViT (Dosovitskiy et al., 2020) pre-trained on ImageNet-22K (Deng et al., 2009) as the transformer encoder. The models are trained for 60,000 iterations on both NYUD (Silberman et al., 2012) and PASCAL (Everingham & Winn, 2012) datasets with batch size 6. We use Adam optimizer with learning rate $2\times10^{-5}$ and $1\times10^{-6}$ of a weight decay with a polynomial learning rate schedule. Following the previous works (Ye & Xu, 2022a;b), we used the same loss and loss scale for each task. The cross-entropy loss was used for semantic segmentation, human parts estimation, and saliency, edge detection. Surface normal prediction and depth estimation used L1 loss.

**Baselines and Model Variants.** For a comprehensive analysis of the proposed DTME-MTL framework, we adopt a typical experimental setup for MTL in our experiments. In Table 1, "Baseline (MT)" refers to a simple multi-task architecture consisting of a shared transformer backbone and basic task-specific decoders. Each decoder comprises one $3 \times 3$ Conv-BN-ReLU block. "Baseline (ST)" has the same structure as "Baseline (MT)" but is trained with only a single task. We assess the proposed DTME-MTL framework by expanding the network from "Baseline (MT)" and measure the performance gains achieved by the proposed methods. "TM" (Token Modulation) signifies the addition of the proposed token modulator to "baseline (MT)," while "TE" (Token Expansion) indicates the incorporation of task-specific tokens onto "Baseline (MT)." Finally, "TM+TE" combines both proposed methods. To show how effectively our approach reduces negative transfer, we also compare it with previous multi-task optimization techniques, though our methods can be used alongside them. We include simple gradient descent (GD), gradient manipulation methods like GradDrop (Chen et al., 2020), MGDA (Sener & Koltun, 2018), PCGrad (Yu et al., 2020), CAGrad (Liu et al., 2021a), IMTL (Liu et al., 2021b), Nash-MTL (Navon et al., 2022), and Aligned-MTL (Senushkin et al., 2023), as well as loss balancing methods such as UW (Kendall et al., 2018), DWA (Liu et al., 2019), and FAMO (Liu et al., 2024). We also compare our results with Recon (Guangyuan et al., 2022) in Appendix D.

**Effectiveness of Token Modulation and Expansion.** We assess the effectiveness of the proposed methods on the NYUD-v2 and PASCAL-Context datasets, with results detailed in Table 1. In the

Table 2: We contrast our methods (TM+TE) with selecting layers based on the degree of conflicts in reversed order (Reversed) and randomly selected layers (Random).

| Model | NYUD-v2 | | | | PASCAL-Context | | | | |
|---|---|---|---|---|---|---|---|---|---|
| | Semseg
mIoU ↑ | Depth
RMSE ↓ | Normal
mErr ↓ | Edge
odsF ↑ | Semseg
mIoU ↑ | Parsing
mIoU ↑ | Saliency
maxF ↑ | Normal
mErr ↓ | Edge
odsF ↑ |
| TM+TE | **38.27** | **0.6370** | **21.64** | **57.90** | **66.18** | **56.29** | **83.21** | **15.26** | **47.00** |
| TM+TE (Random) | 36.88 | 0.6567 | 22.27 | 56.30 | 62.12 | 54.43 | 82.95 | 15.55 | 45.80 |
| TM+TE (Reverse) | 34.71 | 0.6898 | 22.59 | 55.80 | 57.84 | 52.82 | 82.75 | 15.74 | 43.20 |

(a) NYUD-v2        (b) PASCAL-Context

Figure 5: We evaluate the distribution of gradient conflicts by measuring the cosine similarity between task-specific gradients across all shared parameters throughout the optimization process. This is represented as $cos\phi_{ij}$ in (a) for NYUD-v2 and in (b) for PASCAL-Context.

last three rows of the table, we depict the performance gains compared to the two baselines and the increased number of parameters in "$\#Param \uparrow (\%)$". Compared to the Baseline (MT), our methods demonstrate significant performance improvements across all tasks in both datasets. Particularly noteworthy is the substantial increase in multi-task performance achieved with just a 0.2% to 0.3% increase in the total network parameters. Additionally, our approach exhibits nearly identical performance to Baseline (ST) in a multi-task scenario. This implies that reducing negative transfer between tasks can be effectively accomplished by merely integrating introduced token modulators and task-specific tokens, without the need for intricately designed modules.

**Analysis of the Timing of Network Expansion.** In Figure 4, we analyze the performance of each task according to the timing of network expansion using the proposed DTME-MTL. Specifically, the timing for expansion refers to the point at which token modulation and expansion are performed based on calculations of the token space using Singular Value Decomposition and measurement of gradient conflicts. The figure illustrates the performance results when network expansion is conducted at the beginning of training ($0^{th}$) and after each quarter of the entire training process ($i^{th}$ 25% Iter). To ensure fair comparisons, we trained the network using the same number of iterations after the expansion. According to the experimental results, the optimal expansion timing may not align perfectly depending on the task, but overall, it can be observed that performing expansion in the early stages of network training yields better performance.

**Analysis of Gradient Conflicts in Network Parameters.** When using the suggested token modulation and expansion method, unique token spaces emerge for each task, making direct conflict measurement in token space unfeasible. Instead, to evaluate the reduction of conflicts between tasks, we analyze the extent of task-specific gradient interference in the network parameters during the training process. In Figure 5, we divide the angles between task-specific gradients of network parameters into ranges and represent the frequency occurring during the training process. When applying each method to the baseline model, both Token Modulation (TM) and Token Expansion (TM) show a decrease in the ranges where the cosine of the angle between parameter gradients ($\cos \phi_{ij}$) is less than 0, while also showing an increase in the ranges where it is greater than or equal to 0. This indicates that the proposed methods effectively reduce conflicts between tasks and align gradients in the same direction. As a result of reducing conflicts in parameters, it can be observed in Figure 3-(c) that applying both "TM+TE" leads to achieving the lowest multi-task loss.

**Comparing Performance based on Layer Selection Criteria.** In Table 2, we applied token modulation and expansion (TM+TE) to layers with the highest gradient conflicts between tasks. Results are also shown for randomly chosen layers (Random) or layers with the lowest gradient conflicts (Reverse). The network expansion system, using conflict detection, outperforms random selection across all tasks. Particularly, applying TM+TE to layers with severe conflict levels consistently

Table 3: Comparison of multi-task optimization methods on Taskonomy across 11 tasks. Non-converged results are indicated with a dash.

| Task Metric | DE L1 Dist.↓ | DZ L1 Dist.↓ | EO L1 Dist.↓ | ET L1 Dist.↓ | Key2D L1 Dist.↓ | Key3D L1 Dist.↓ | N L1 Dist. | PC RMSE↓ | R L1 Dist.↓ | S2D L1 Dist.↓ | S25D L1 Dist.↓ | $\triangle_m$↑(%) |
|---|---|---|---|---|---|---|---|---|---|---|---|---|
| ST | 0.0199 | 0.0195 | 0.1085 | 0.1714 | 0.1633 | 0.0872 | 0.2715 | 0.7586 | 0.1503 | 0.1742 | 0.1504 | 0.00 |
| GD | 0.0187 | 0.0188 | 0.1301 | 0.1757 | 0.1733 | 0.0942 | 0.3076 | 0.7991 | 0.1826 | 0.1902 | 0.1652 | - 7.83 |
| GradDrop | 0.0315 | 0.0242 | 0.1390 | 0.1776 | 0.1778 | 0.0976 | 0.4564 | 0.8644 | 0.2088 | 0.1995 | 0.1752 | - 26.11 |
| MGDA | - | - | - | - | - | - | - | - | - | - | - | - |
| UW | 0.0190 | 0.0190 | 0.1308 | 0.1758 | 0.1734 | 0.0945 | 0.3109 | 0.8009 | 0.1840 | 0.1906 | 0.1657 | - 8.43 |
| DWA | 0.0186 | 0.0187 | 0.1294 | 0.1759 | 0.1735 | 0.0938 | 0.2788 | 0.7943 | 0.1805 | 0.1902 | 0.1640 | - 6.45 |
| PCGrad | 0.0217 | 0.0192 | 0.1298 | 0.1775 | 0.1714 | 0.0939 | 0.2856 | 0.7985 | 0.1817 | 0.1927 | 0.1595 | - 8.29 |
| CAGrad | 0.0219 | 0.0203 | 0.1314 | 0.1800 | 0.1665 | 0.0932 | 0.3039 | 0.8121 | 0.1874 | 0.1953 | 0.1673 | - 10.57 |
| IMTL | 0.0210 | 0.0192 | 0.1282 | 0.1772 | 0.1719 | 0.0936 | 0.2468 | 0.7784 | 0.1734 | 0.1943 | 0.1647 | - 6.17 |
| Align-MTL | 0.0189 | 0.0193 | 0.1254 | **0.1728** | **0.1664** | 0.0914 | 0.3524 | 0.8640 | 0.1938 | 0.1889 | 0.1582 | - 9.41 |
| Nash-MTL | 0.0201 | 0.0184 | 0.1248 | 0.1764 | 0.1701 | 0.0921 | 0.2658 | 0.7793 | 0.1706 | 0.1914 | 0.1624 | - 5.01 |
| FAMO | 0.0188 | 0.0188 | 0.1300 | 0.1758 | 0.1733 | 0.0942 | 0.3058 | 0.7986 | 0.1826 | 0.1904 | 0.1654 | - 7.87 |
| DTME-MTL | **0.0150** | **0.0154** | **0.1193** | 0.1733 | 0.1668 | **0.0891** | 0.2038 | 0.7373 | 0.1567 | 0.1773 | 0.1517 | + 4.67 |

Table 4: Adaptation of DTME-MTL to other MTL methods: We evaluate performance on NYUD-v2 (*left*) and PASCAL-Context (*right*). Existing studies are divided into CNN-based and transformer-based models. The best results are shown in **bold**, and the second-best are underlined.

| Task Metric | Semseg mIoU↑ | Depth RMSE↓ | Normal mErr↓ | Edge odsF↑ |
|---|---|---|---|---|
| Cross-Stitch | 36.34 | 0.6290 | 20.88 | 76.38 |
| PAP | 36.72 | 0.6178 | 20.82 | 76.42 |
| PSD | 36.69 | 0.6246 | 20.87 | 76.42 |
| PAD-Net | 36.61 | 0.6270 | 20.85 | 76.38 |
| MTI-Net | 45.97 | 0.5365 | 20.27 | 77.86 |
| ATRC | 46.33 | 0.5363 | 20.18 | 77.94 |
| MTformer | 50.04 | 0.490 | - | - |
| InvPT | 53.56 | 0.5183 | 18.81 | 78.10 |
| + DTME-MTL | 54.38 | **0.5020** | 18.51 | 78.20 |
| Taskprompter | 55.30 | 0.5152 | 18.47 | 78.20 |
| + DTME-MTL | **56.36** | 0.5122 | **18.38** | **78.40** |

| Task Metric | Semseg mIoU↑ | Parsing mIoU↑ | Saliency maxF↑ | Normal mErr↓ | Edge odsF↑ |
|---|---|---|---|---|---|
| ASTMT | 68.00 | 61.10 | 65.70 | 14.70 | 72.40 |
| PAD-Net | 53.60 | 59.60 | 65.80 | 15.30 | 72.50 |
| MTI-Net | 61.70 | 60.18 | 84.78 | 14.23 | 70.80 |
| ATRC | 62.69 | 59.42 | 84.70 | 14.20 | 70.96 |
| ATRC-ASPP | 63.60 | 60.23 | 83.91 | 14.30 | 70.86 |
| ATRC-BMTAS | 67.67 | 62.93 | 82.29 | 14.24 | 72.42 |
| MTformer | 73.51 | 64.26 | 67.24 | - | - |
| InvPT | 79.03 | 67.61 | 84.81 | 14.15 | 73.00 |
| + DTME-MTL | **81.91** | **71.13** | **84.96** | 13.73 | **73.80** |
| Taskprompter | 80.89 | 68.89 | 84.83 | 13.72 | 73.50 |
| + DTME-MTL | 81.01 | 69.08 | 84.75 | **13.65** | 73.60 |

outperforms its application in layers with lower conflict levels, validating the effectiveness of the proposed expansion strategy.

**Comparison with Multi-Task Optimization.** In Table 3, we compare DTME-MTL with previous multi-task optimization approaches to demonstrate its effectiveness in reducing negative transfer between tasks on the Taskonomy benchmark using ViT-B. DTME-MTL achieves the best multi-task performance, improving each task by an average of $4.67\%$ with only a $0.118\%$ increase in the number of parameters. Although DTME-MTL introduces additional parameters to address negative transfer, making direct comparisons with optimization methods less straightforward, it consistently improves multi-task performance. However, using more task-specific parameters does not always yield better results, as Recon (Guangyuan et al., 2022) shows poor performance with the vision transformer on NYUD-v2 (Table 10).

**Adapting to Multi-Task Architectures.** In Table 4, we compare DTME-MTL with leading multi-task architectures on the NYUD-v2 and PASCAL-Context datasets. We evaluate its multi-task performance against CNN-based methods such as Cross-Stitch (Misra et al., 2016), ASTMT (Maninis et al., 2019), PAP (Zhang et al., 2019), PSD (Zhou et al., 2020), PAD-Net (Xu et al., 2018), MTI-Net (Vandenhende et al., 2020), ATRC (Brüggemann et al., 2021), and transformer-based approaches like MTformer (Xu et al., 2022), InvPT (Ye & Xu, 2022a), and TaskPrompter (Ye & Xu, 2022b). Our method is compatible with any transformer-based multi-task architecture, enabling us to assess its effectiveness by integrating it into two leading models: InvPT and TaskPrompter. DTME-MTL seamlessly enhances these architectures, significantly boosting performance with only a minimal increase in parameters — just $0.048\%$ for InvPT and $0.046\%$ for TaskPrompter.

## 6 CONCLUSION

This paper presents Dynamic Token Modulation and Expansion for Multi-Task Learning (DTME-MTL), a novel approach aimed at improving transformer-based multi-task architectures by addressing gradient conflicts among tasks. We categorize conflicts between tasks based on whether they occur within token range space or null space. Using this categorization, we adaptively apply token modulation and expansion to mitigate these conflicts. The proposed system effectively reduces task conflicts, leading to enhanced multi-task performance. Our method can be easily integrated into different transformer-based multi-task architectures with only a small number of additional parameters, achieving superior performance on various multi-task benchmarks.

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

# A THEORETICAL ANALYSIS

## A.1 PROOF OF THEOREM 1

**Theorem 1.** *Optimizing the token modulators $\{\mathcal{M}_i\}_{i=1}^{\mathcal{K}}$ reduces gradient conflicts in the row space of $\tilde{\mathcal{T}}_s$ and leads to a reduction in the multi-task loss.*

*Proof.* Let the loss function $\mathcal{L}_i$ be a function of the shared parameters $\Theta_s$, the token modulator $\mathcal{M}_i$, and the input data $\mathcal{X}^t$. Since transformers convert input data into tokens, we consider the loss to be a function of the input token $\mathcal{T}_{in}$ rather than $\mathcal{X}^t$. In cases where the input token $\mathcal{T}_{in}^t$ spans the row space of $\tilde{\mathcal{T}}_s$, this can be expressed as follows:

$$\mathcal{U}_{\mathcal{N}}\mathcal{U}_{\mathcal{N}}^{T}\nabla_{\mathcal{T}_{in}}\mathcal{L}_i(\Theta_s^t, \mathcal{M}_i^t, \mathcal{T}_{in}^t) \simeq 0 \tag{5}$$

Since the row space and null space are perpendicular to each other, with their dimensions summing to the entire space, the following holds according to eq. (5):

$$\sum_{i=1}^{\mathcal{K}} \nabla_{\mathcal{T}_{in}^t}\mathcal{L}_i = \sum_{i=1}^{\mathcal{K}} (\mathcal{U}_{\mathcal{R}}\mathcal{U}_{\mathcal{R}}^{T} + \mathcal{U}_{\mathcal{N}}\mathcal{U}_{\mathcal{N}}^{T})\nabla_{\mathcal{T}_{in}^t}\mathcal{L}_i \simeq \sum_{i=1}^{\mathcal{K}} (\mathcal{U}_{\mathcal{R}}\mathcal{U}_{\mathcal{R}}^{T})\nabla_{\mathcal{T}_{in}^t}\mathcal{L}_i \tag{6}$$

Let the token modulator $\mathcal{M}_i$ be a $d \times d$ matrix that manipulates the input token $\mathcal{T}_{in}$.

$$\sum_{i=1}^{\mathcal{K}} \nabla_{\mathcal{T}_{in}^t}\mathcal{L}_i = \sum_{i=1}^{\mathcal{K}} (\mathcal{U}_{\mathcal{R}}\mathcal{M}_i^t)(\mathcal{U}_{\mathcal{R}}\mathcal{M}_i^t)^T \cdot \nabla_{\mathcal{M}_i^t}\mathcal{L}_i \cdot \nabla_{\mathcal{T}_{in}^t}\mathcal{M}_i^t \tag{7}$$

The total multi-task loss can be represented using a Taylor expansion. Assuming $\eta \ll 1$, we can ignore the second-order terms of $\eta$:

$$\sum_{i=1}^{\mathcal{K}} \mathcal{L}_i(\Theta_s^{t+1}, \mathcal{M}_i^{t+1}, \mathcal{T}_s^t) = \sum_{i=1}^{\mathcal{K}} \mathcal{L}_i(\Theta_s^t, \mathcal{M}_i^t, \mathcal{T}_s^t) + \sum_{i=1}^{\mathcal{K}} \nabla_{\Theta_s^t}\mathcal{L}_i(\Theta_s^t, \mathcal{M}_i^t, \mathcal{T}_s^t)(\Theta_s^{t+1} - \Theta_s^t) \tag{8}$$

$$+ \sum_{i=1}^{\mathcal{K}} \nabla_{\mathcal{M}_i^t}\mathcal{L}_i(\Theta_s^t, \mathcal{M}_i^t, \mathcal{T}_s^t)(\mathcal{M}_i^{t+1} - \mathcal{M}_i^t) \tag{9}$$

$$= \sum_{i=1}^{\mathcal{K}} \mathcal{L}_i(\Theta_s^t, \mathcal{M}_i^t, \mathcal{T}_s^t) - \eta|\sum_{i=1}^{\mathcal{K}} \nabla_{\Theta_s^t}\mathcal{L}_i(\Theta_s^t, \mathcal{M}_i^t, \mathcal{T}_s^t)|^2 \tag{10}$$

$$- \eta \sum_{i=1}^{\mathcal{K}} |\nabla_{\mathcal{M}_i^t}\mathcal{L}_i(\Theta_s^t, \mathcal{M}_i^t, \mathcal{T}_s^t)|^2 \tag{11}$$

By optimizing the modulator $\mathcal{M}_i^t$ so that $|\nabla_{\mathcal{M}_i^t}\mathcal{L}_i(\Theta_s^t, \mathcal{M}_i^t, \mathcal{T}_{in}^t)|$ approaches zero for each task $i = 1, 2, \ldots, \mathcal{K}$, we can alleviate gradient conflicts in the row space of $\tilde{\mathcal{T}}_s$ (as eq. (7) also approaches zero) and reduce the overall multi-task loss, since eq. (11) is always greater than or equal to zero. $\square$

## A.2 PROOF OF THEOREM 2

**Theorem 2.** *Token expansion using $\{\mathcal{T}_i\}_{i=1}^{\mathcal{K}}$ alleviates the increase in multi-task loss caused by gradient conflicts in the null space of $\tilde{\mathcal{T}}_s$.*

*Proof.* Let the loss function $\mathcal{L}_i$ be a function of the shared parameters $\Theta_s^t$, the task-specific token $\mathcal{T}_i^t$, and the input data $\mathcal{X}^t$. Similarly, since transformers convert input data into tokens, we consider the loss as a function of the input token $\mathcal{T}_{in}^t$ rather than $\mathcal{X}^t$. In the case where the input token $\mathcal{T}_{in}^t$ spans the null space of $\tilde{\mathcal{T}}_s$, this can be expressed as follows:

$$\sum_{i=1}^{\mathcal{K}} \mathcal{U}_{\mathcal{R}}\mathcal{U}_{\mathcal{R}}^{T}\nabla_{\mathcal{T}_{in}^t}\mathcal{L}_i(\Theta_s^t, \mathcal{T}_{in}^t, \mathcal{T}_i^t) \simeq 0 \tag{12}$$

The derivative of the task-specific loss $\mathcal{L}_i$ with respect to the expanded token, including the input token $\mathcal{T}_{in}^t$ and the learnable task-specific tokens $\mathcal{T}_i^t$, is given as follows:

$$\sum_{i=1}^{\mathcal{K}} \nabla_{\{\mathcal{T}_{in}^t, \mathcal{T}_i^t\}} \mathcal{L}_i \tag{13}$$

$$= \sum_{i=1}^{\mathcal{K}} \left( \begin{bmatrix} \mathcal{U}_{\mathcal{R}} & 0_{d \times \mathcal{K}} \\ 0_{\mathcal{K} \times d} & \mathcal{U}_{\mathcal{R},i} \end{bmatrix} \begin{bmatrix} \mathcal{U}_{\mathcal{R}} & 0_{d \times \mathcal{K}} \\ 0_{\mathcal{K} \times d} & \mathcal{U}_{\mathcal{R},i} \end{bmatrix}^T + \begin{bmatrix} \mathcal{U}_{\mathcal{N}} & 0_{d \times \mathcal{K}} \\ 0_{\mathcal{K} \times d} & 0_{\mathcal{K} \times \mathcal{K}} \end{bmatrix} \begin{bmatrix} \mathcal{U}_{\mathcal{N}} & 0_{d \times \mathcal{K}} \\ 0_{\mathcal{K} \times d} & 0_{\mathcal{K} \times \mathcal{K}} \end{bmatrix}^T \right) \begin{bmatrix} \nabla_{\mathcal{T}_{in}^t} \mathcal{L}_i \\ \nabla_{\mathcal{T}_i^t} \mathcal{L}_i \end{bmatrix} \tag{14}$$

$$= \sum_{i=1}^{\mathcal{K}} \begin{bmatrix} \mathcal{U}_{\mathcal{R}} \mathcal{U}_{\mathcal{R}}^T + \mathcal{U}_{\mathcal{N}} \mathcal{U}_{\mathcal{N}}^T & 0_{d \times \mathcal{K}} \\ 0_{\mathcal{K} \times d} & \mathcal{U}_{\mathcal{R},i} \mathcal{U}_{\mathcal{R},i}^T \end{bmatrix} \begin{bmatrix} \nabla_{\mathcal{T}_{in}^t} \mathcal{L}_i \\ \nabla_{\mathcal{T}_i^t} \mathcal{L}_i \end{bmatrix} \tag{15}$$

$$\simeq \sum_{i=1}^{\mathcal{K}} \begin{bmatrix} \mathcal{U}_{\mathcal{N}} \mathcal{U}_{\mathcal{N}}^T & 0_{d \times \mathcal{K}} \\ 0_{\mathcal{K} \times d} & \mathcal{U}_{\mathcal{R},i} \mathcal{U}_{\mathcal{R},i}^T \end{bmatrix} \begin{bmatrix} \nabla_{\mathcal{T}_{in}^t} \mathcal{L}_i \\ \nabla_{\mathcal{T}_i^t} \mathcal{L}_i \end{bmatrix} \tag{16}$$

$$= \sum_{i=1}^{\mathcal{K}} \begin{bmatrix} (\mathcal{U}_{\mathcal{N}} \mathcal{U}_{\mathcal{N}}^T) \nabla_{\mathcal{T}_{in}^t} \mathcal{L}_i \\ (\mathcal{U}_{\mathcal{R},i} \mathcal{U}_{\mathcal{R},i}^T) \nabla_{\mathcal{T}_i^t} \mathcal{L}_i \end{bmatrix} \tag{17}$$

The total multi-task loss can be expressed as follows:

$$\mathcal{L}_i(\Theta_s^{t+1}, \mathcal{T}_{in}^{t+1}, \mathcal{T}_i^{t+1}) = \mathcal{L}_i(\Theta_{in}^t, \mathcal{T}_s^t, \mathcal{T}_i^t) + \nabla_{\Theta_s^t} \mathcal{L}_i(\Theta_s^t, \mathcal{T}_s^t, \mathcal{T}_i^t)(\Theta_s^{t+1} - \Theta_s^t) \tag{18}$$

$$+ \nabla_{\mathcal{T}_{in}^t} \mathcal{L}_i(\Theta_s^t, \mathcal{T}_s^t, \mathcal{T}_i^t)(\mathcal{T}_{in}^{t+1} - \mathcal{T}_{in}^t) \tag{19}$$

$$+ \nabla_{\mathcal{T}_i^t} \mathcal{L}_i(\Theta_s^t, \mathcal{T}_s^t, \mathcal{T}_i^t)(\mathcal{T}_i^{t+1} - \mathcal{T}_i^t) \tag{20}$$

$$= \mathcal{L}_i(\Theta_s^t, \mathcal{T}_{in}^t, \mathcal{T}_i^t) - \eta \nabla_{\Theta_s^t} \mathcal{L}_i(\Theta_s^t, \mathcal{T}_s^t, \mathcal{T}_i^t) \cdot \sum_{i=1}^{\mathcal{K}} \nabla_{\Theta_s^t} \mathcal{L}_i(\Theta_s^t, \mathcal{T}_{in}^t, \mathcal{T}_i^t) \tag{21}$$

$$- \eta (\mathcal{U}_{\mathcal{N}} \mathcal{U}_{\mathcal{N}}^T) \nabla_{\mathcal{T}_{in}^t} \mathcal{L}_i(\Theta_s^t, \mathcal{T}_{in}^t, \mathcal{T}_i^t) \cdot \sum_{i=1}^{\mathcal{K}} (\mathcal{U}_{\mathcal{N}} \mathcal{U}_{\mathcal{N}}^T) \nabla_{\mathcal{T}_{in}^t} \mathcal{L}_i(\Theta_s^t, \mathcal{T}_{in}^t, \mathcal{T}_i^t) \tag{22}$$

$$- \eta (\mathcal{U}_{\mathcal{R},i} \mathcal{U}_{\mathcal{R},i}^T) \nabla_{\mathcal{T}_i^t} \mathcal{L}_i(\Theta_s^t, \mathcal{T}_{in}^t, \mathcal{T}_i^t) \cdot (\mathcal{U}_{\mathcal{R},i} \mathcal{U}_{\mathcal{R},i}^T) \nabla_{\mathcal{T}_i^t} \mathcal{L}_i(\Theta_s^t, \mathcal{T}_{in}^t, \mathcal{T}_i^t) \tag{23}$$

The increase in multi-task loss caused by gradient conflicts in the null space (as described in eq. (22)) cannot be reduced since the shared token $\mathcal{T}_{in}^t$ is not a learnable parameter. Instead, task-specific tokens $\mathcal{T}_i^t$ can be added to mitigate the increase in multi-task loss due to null space gradient conflicts by optimizing the learnable parameters $\{\mathcal{T}_i\}_{i=1}^{\mathcal{K}}$ as described in eq. (23).

$\square$

## B ADDITIONAL RELATED WORKS

**Multi-Task Architectures.** Various multi-task architectures can be categorized based on how the parameters or features of the sharing network are distributed among tasks. The widely used shared trunk structure comprises a common encoder shared by multiple tasks and a dedicated decoder for each task (Dai et al., 2016; Ma et al., 2018; Simonyan & Zisserman, 2014; Zhang et al., 2014). A tree-like architecture, with multiple division points for each task group, offers a more generalized structure (Lu et al., 2017; Vandenhende et al., 2019; Bruggemann et al., 2020; Guo et al., 2020). The cross-talk architecture employs separate symmetrical networks for each task, utilizing feature exchange between layers at the same depth for information sharing between tasks (Gao et al., 2019; Xu et al., 2018). The prediction distillation model (Eigen & Fergus, 2015; Xu et al., 2018; Vandenhende et al., 2020; Zhang et al., 2019) incorporates cross-task interactions at the end of the shared encoder, while the task switching network (Sun et al., 2021; Sinha et al., 2018; Fernando et al., 2017; Maninis et al., 2019) changes network parameters depending on the task.

## C  EXPERIMENTAL SETTINGS

**Datasets:** These datasets contain different kinds of vision tasks. NYUD-v2 contains 4 vision tasks: Our evaluation is based on depth estimation, semantic segmentation, surface normal prediction, and edge detection. PASCAL-Context contains 5 tasks: We evaluate semantic segmentation, human parts estimation, saliency estimation, surface normal prediction, and edge detection. We used 11 tasks for Taskonomy: We evaluate Depth Euclidean (DE), Depth Zbuffer (DZ), Edge Texture (ET), Keypoints 2D (Key2D), Keypoints 3D (Key3D), Normal (N), Principal Curvature (PC), Reshading (R), Segment Unsup 2d (S2D), and Segment Unsup 2.5D (S25D).

**Evaluation.** For semantic segmentation, we utilized mean Intersection over Union (mIoU). Surface normal prediction's performance was measured by calculating the mean angle distances between the predicted output and ground truth. To evaluate the depth estimation task, we used Root Mean Squared Error (RMSE). For saliency estimation and human part segmentation, we employed mean Intersection over Union (mIoU). For edge detection, we used optimal-dataset-scale-F-measure (odsF). For Taskonomy, we adopt

## D  ADDITIONAL EXPERIMENTS

**Comparison with Multi-Task Optimization.** In Tables 5 to 7, we further evaluate the proposed DTME-MTL against previous multi-task optimization approaches using different backbone sizes. Our method demonstrates significant improvements in multi-task performance with minimal increases in parameters. Specifically, DTME-MTL results in a parameter increase of $0.089\%$ for ViT-L, $0.23\%$ for ViT-S, and $0.46\%$ for ViT-T.

Table 5: Comparison with multi-task optimization approaches on Taskonomy across 11 different tasks with ViT-L. Non-converged results are indicated with a dash.

| Task Metric | DE L1 Dist. ↓ | DZ L1 Dist. ↓ | EO L1 Dist. ↓ | ET L1 Dist. ↓ | Key2D L1 Dist. ↓ | Key3D L1 Dist. ↓ | N L1 Dist. | PC RMSE ↓ | R L1 Dist. ↓ | S2D L1 Dist. ↓ | S25D L1 Dist. ↓ | $\triangle_m$ ↑ (%) |
|---|---|---|---|---|---|---|---|---|---|---|---|---|
| ST | 0.0141 | 0.0146 | 0.0992 | 0.1716 | 0.1631 | 0.0801 | 0.2133 | 0.7134 | 0.1342 | 0.1688 | 0.1419 | 0.00 |
| GD | 0.0153 | 0.0156 | 0.1196 | 0.1757 | 0.1729 | 0.0896 | 0.2215 | 0.7451 | 0.1576 | 0.1826 | 0.1537 | -8.92 |
| GradDrop | 0.0170 | 0.0195 | 0.1235 | 0.1757 | 0.1753 | 0.0909 | 0.2818 | 0.7679 | 0.1663 | 0.1916 | 0.1543 | -17.07 |
| MGDA | - | - | - | - | - | - | - | - | - | - | - | - |
| UW | 0.0152 | 0.0155 | 0.1195 | 0.1755 | 0.1728 | 0.0897 | 0.2356 | 0.7436 | 0.1569 | 0.1830 | 0.1538 | -9.36 |
| DWA | 0.0153 | 0.0156 | 0.1197 | 0.1757 | 0.1730 | 0.0897 | 0.2214 | 0.7441 | 0.1576 | 0.1827 | 0.1537 | -8.96 |
| PCGrad | 0.0152 | 0.0156 | 0.1192 | 0.1749 | 0.1699 | 0.0893 | 0.2310 | 0.7475 | 0.1577 | 0.1825 | 0.1480 | -8.63 |
| CAGrad | 0.0155 | 0.0156 | 0.1175 | 0.1756 | 0.1649 | 0.0860 | 0.2421 | 0.7544 | 0.1591 | 0.1854 | 0.1554 | -9.32 |
| IMTL | 0.0151 | 0.0156 | 0.1194 | 0.1755 | 0.1726 | 0.0895 | 0.2199 | 0.7432 | 0.1569 | 0.1824 | 0.1533 | -8.57 |
| Align-MTL | 0.0150 | 0.0155 | 0.1136 | 0.1733 | 0.1633 | 0.0862 | 0.2512 | 0.8029 | 0.1643 | 0.1803 | 0.1445 | -8.78 |
| Nash-MTL | 0.0151 | 0.0154 | 0.1138 | 0.1732 | 0.1644 | 0.0863 | 0.2507 | 0.7656 | 0.1544 | 0.1833 | 0.1452 | -7.95 |
| FAMO | 0.0153 | 0.0157 | 0.1196 | 0.1757 | 0.1730 | 0.0897 | 0.2221 | 0.7444 | 0.1575 | 0.1830 | 0.1534 | -8.99 |
| DTME-MTL | 0.0127 | 0.0130 | 0.1088 | 0.1731 | 0.1665 | 0.0852 | 0.1654 | 0.6890 | 0.1389 | 0.1661 | 0.1404 | +2.41 |

Table 6: Comparison with multi-task optimization approaches on Taskonomy across 11 different tasks with ViT-S. Non-converged results are indicated with a dash.

| Task Metric | DE L1 Dist.↓ | DZ L1 Dist.↓ | EO L1 Dist.↓ | ET L1 Dist.↓ | Key2D L1 Dist.↓ | Key3D L1 Dist.↓ | N L1 Dist. | PC RMSE↓ | R L1 Dist.↓ | S2D L1 Dist.↓ | S25D L1 Dist.↓ | $\triangle_m \uparrow$(%) |
|---|---|---|---|---|---|---|---|---|---|---|---|---|
| ST 0.0255 | 0.0255 | 0.1285 | 0.1727 | 0.1653 | 0.0918 | 0.3973 | 0.8562 | 0.1864 | 0.1824 | 0.1647 | 0.00 | |
| GD | 0.0244 | 0.0243 | 0.1501 | 0.1778 | 0.1844 | 0.1009 | 0.4105 | 0.9087 | 0.2325 | 0.2032 | 0.1822 | -8.04 |
| GradDrop | 0.0253 | 0.0253 | 0.1533 | 0.1785 | 0.1865 | 0.1021 | 0.4399 | 0.9246 | 0.2408 | 0.2063 | 0.1791 | -10.42 |
| MGDA | - | - | - | - | - | - | - | - | - | - | - | - |
| UW | 0.0242 | 0.0242 | 0.1498 | 0.1778 | 0.1847 | 0.1007 | 0.4064 | 0.9079 | 0.2312 | 0.2033 | 0.1822 | -7.74 |
| DWA | 0.0242 | 0.0242 | 0.1500 | 0.1778 | 0.1844 | 0.1008 | 0.4097 | 0.9071 | 0.2316 | 0.2032 | 0.1822 | -7.84 |
| PCGrad | 0.0248 | 0.0248 | 0.1501 | 0.1755 | 0.1761 | 0.1001 | 0.4306 | 0.9181 | 0.2371 | 0.2023 | 0.1772 | -8.12 |
| CAGrad | 0.0254 | 0.0255 | 0.1516 | 0.1738 | 0.1698 | 0.0983 | 0.4535 | 0.9282 | 0.2442 | 0.2068 | 0.1849 | -9.74 |
| IMTL | 0.0236 | 0.0237 | 0.1456 | 0.1756 | 0.1760 | 0.0988 | 0.4151 | 0.9055 | 0.2222 | 0.2010 | 0.1794 | -5.74 |
| Align-MTL | 0.0266 | 0.0264 | 0.1499 | 0.1736 | 0.1700 | 0.0986 | 0.4659 | 0.9868 | 0.2604 | 0.2030 | 0.1780 | -11.51 |
| Nash-MTL | 0.0235 | 0.0235 | 0.1432 | 0.1745 | 0.1718 | 0.0975 | 0.4230 | 0.9225 | 0.2268 | 0.1985 | 0.1775 | -5.41 |
| FAMO | 0.0243 | 0.0243 | 0.1499 | 0.1778 | 0.1846 | 0.1008 | 0.3841 | 0.9080 | 0.2321 | 0.2027 | 0.1816 | -7.31 |
| DTME-MTL | 0.0196 | 0.0200 | 0.1372 | 0.1754 | 0.1712 | 0.0958 | 0.3129 | 0.8333 | 0.1955 | 0.1907 | 0.1698 | +3.62 |

Table 7: Comparison with multi-task optimization approaches on Taskonomy across 11 different tasks with ViT-T. Non-converged results are indicated with a dash.

| Task Metric | DE L1 Dist.↓ | DZ L1 Dist.↓ | EO L1 Dist.↓ | ET L1 Dist.↓ | Key2D L1 Dist.↓ | Key3D L1 Dist.↓ | N L1 Dist. | PC RMSE↓ | R L1 Dist.↓ | S2D L1 Dist.↓ | S25D L1 Dist.↓ | $\triangle_m \uparrow$(%) |
|---|---|---|---|---|---|---|---|---|---|---|---|---|
| ST | 0.0250 | 0.0256 | 0.1388 | 0.1755 | 0.1670 | 0.0958 | 0.3856 | 0.9066 | 0.2132 | 0.1878 | 0.1722 | 0.00 |
| GD | 0.0266 | 0.0278 | 0.1593 | 0.1794 | 0.1865 | 0.1047 | 0.4752 | 0.9467 | 0.2568 | 0.2081 | 0.1897 | -11.10 |
| GradDrop | 0.0276 | 0.0284 | 0.1624 | 0.1807 | 0.1884 | 0.1064 | 0.4741 | 0.9611 | 0.2658 | 0.2108 | 0.1860 | -12.67 |
| MGDA | - | - | - | - | - | - | - | - | - | - | - | - |
| UW | 0.0266 | 0.0277 | 0.1593 | 0.1795 | 0.1865 | 0.1045 | 0.4757 | 0.9466 | 0.2567 | 0.2080 | 0.1896 | -11.07 |
| DWA | 0.0266 | 0.0274 | 0.1593 | 0.1794 | 0.1866 | 0.1045 | 0.4743 | 0.9465 | 0.2567 | 0.2080 | 0.1897 | -10.95 |
| PCGrad | 0.0273 | 0.0285 | 0.1596 | 0.1768 | 0.1807 | 0.1043 | 0.4785 | 0.9689 | 0.2644 | 0.2080 | 0.1854 | -11.55 |
| CAGrad | 0.0290 | 0.0305 | 0.1641 | 0.1747 | 0.1731 | 0.1051 | 0.4884 | 0.9870 | 0.2828 | 0.2136 | 0.1945 | -14.64 |
| IMTL | 0.0263 | 0.0272 | 0.1558 | 0.1772 | 0.1810 | 0.1025 | 0.4730 | 0.9525 | 0.2458 | 0.2065 | 0.1868 | -9.24 |
| Align-MTL | - | - | - | - | - | - | - | - | - | - | - | - |
| Nash-MTL | 0.0261 | 0.0270 | 0.1536 | 0.1762 | 0.1766 | 0.1017 | 0.4590 | 0.9649 | 0.2496 | 0.2039 | 0.1846 | -8.28 |
| FAMO | 0.0266 | 0.0275 | 0.1592 | 0.1795 | 0.1865 | 0.1047 | 0.4746 | 0.9466 | 0.2566 | 0.2080 | 0.1898 | -10.97 |
| DTME-MTL | 0.0236 | 0.0241 | 0.1494 | 0.1765 | 0.1790 | 0.0998 | 0.4138 | 0.8921 | 0.2290 | 0.1959 | 0.1824 | -2.88 |

**Analysis on the Modulator Configuration.** In Table 8, we show the performance difference based on the configuration of the token modulators. Specifically, we compared the outcomes obtained when employing affine transformation and batch normalization, which could be considered as the most common and straightforward approaches. Through experiments, we find that affine transformations consistently exhibit better performance across all tasks compared to batch normalization layers used as modulators for both datasets.

Table 8: We compare task performance based on the configuration of the modulator. Specifically, we compare the performance of tasks using an affine transformation against those using a batch normalization layer as configurations for the modulator.

| Model | NYUD-v2 | | | | PASCAL-Context | | | | |
|---|---|---|---|---|---|---|---|---|---|
| | Semseg mIoU↑ | Depth RMSE↓ | Normal mErr↓ | Edge odsF↑ | Semseg mIoU↑ | Parsing mIoU↑ | Saliency maxF↑ | Normal mErr↓ | Edge odsF↑ |
| TM+TE (Affine) | **38.27** | **0.6370** | **21.64** | **57.90** | **66.18** | **56.29** | **83.21** | 15.26 | **47.00** |
| TM+TE (Batch Norm) | 37.42 | 0.6550 | 23.16 | 56.10 | 60.80 | 53.29 | 82.59 | 15.73 | 44.90 |

**Analyzing Performance Differences with Backbone Network Freezing.** In Table 9, we examine the performance variation based on whether we freeze the existing backbone network components when training the expanded network after implementing the proposed dynamic token modulation and expansion. The results indicate that training networks without freezing the existing backbone network components leads to significantly better performance compared to training networks with freezing. We guess that allowing modifications to the learned token space after expansion helps the network to dynamically partition the token space for each task.

**Influence of $r$ on SVD Approximation.** In Figure 6, we illustrate how the proportion of total variance $r$ impacts the approximation of a token's range and null space. We assess the performance of tasks across five values of $r$ (1, 10, 100, 500, 1000). Our results suggest that the value of $r$ has minimal impact on task performance, implying that there is less need for extensive tuning of the $r$ parameter to optimize performance. In our other experiments, we chose $r$ as 100 for training.

**The Impact of the Number of Layers Expanded by DTME-MTL.** DTME-MTL enables the expansion of a specified number of layers and selects those with the highest degree of gradient conflicts. In Figure 7, we illustrate how the performance of tasks is influenced by the number of expanded layers. Specifically, we use the ratio of expanded layers to the total number of layers as the x-axis in the graphs. The findings suggest that ratios between roughly 0.25 and 0.5 show improved performance trends across various tasks, while still maintaining adequate parameter efficiency.

Table 9: We assess task performance by comparing scenarios where we freeze the backbone network after expansion (w/ Freeze) and where we don't (w/o Freeze).

| Model | NYUD-v2 | | | | PASCAL-Context | | | | |
|---|---|---|---|---|---|---|---|---|---|
| | Semseg mIoU ↑ | Depth RMSE ↓ | Normal mErr ↓ | Edge odsF ↑ | Semseg mIoU ↑ | Parsing mIoU ↑ | Saliency maxF ↑ | Normal mErr ↓ | Edge odsF ↑ |
| TM+TE (w/ Freeze) | 34.80 | 0.6730 | 22.48 | 56.00 | 58.34 | 52.96 | 82.86 | 15.63 | 43.20 |
| TM+TE (w/o Freeze) | **38.27** | **0.6370** | **21.64** | **57.90** | **66.18** | **56.29** | **83.21** | **15.26** | **47.00** |

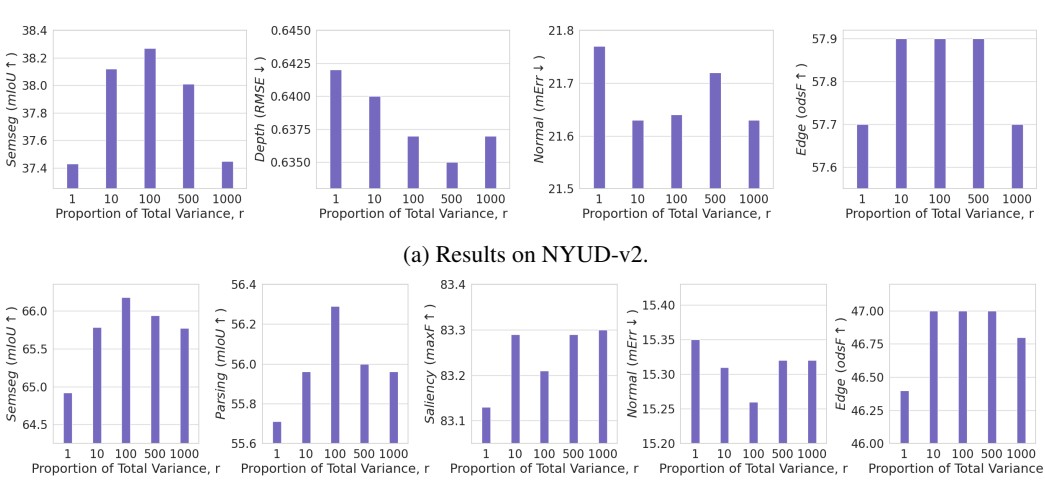

(a) Results on NYUD-v2.

(b) Results on PASCAL-Context.

Figure 6: We assess the performance of tasks based on the proportion of total variance $r$. The results are displayed for both (a) NYUD-v2 and (b) PASCAL-Context.

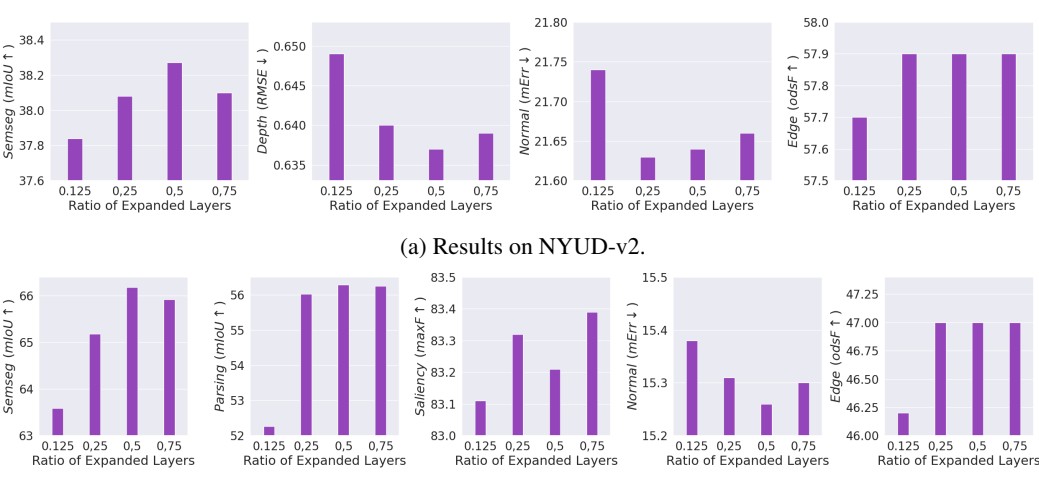

(a) Results on NYUD-v2.

(b) Results on PASCAL-Context.

Figure 7: The performance of tasks based on the ratio of the number of expanded layers to the total number of layers. The results are displayed for both (a) NYUD-v2 and (b) PASCAL-Context.

Table 10: Comparison with Recon on NYUD

| Method | Semseg mIoU ↑ | Depth RMSE ↓ | Normal mErr ↓ | Edge odsF ↑ | #Param ↑ (%) |
|---|---|---|---|---|---|
| Joint | 34.13 | 0.673 | 22.51 | 56.38 | 0.0 |
| Recon | 31.92 | 0.693 | 23.35 | 52.80 | 23.34 |
| Ours | **38.27** | **0.6370** | **21.64** | **57.90** | 0.24 |

