# OpenReview forum: "Dynamic Token Modulation and Expansion for Multi-Task Learning"
_ICLR.cc/2025/Conference — ICLR 2025 Conference Withdrawn Submission_

### Official Review · Reviewer_M8YR · 2024-10-25

**Soundness:** 2
**Presentation:** 2
**Contribution:** 2
**Rating:** 3
**Confidence:** 4

**Summary:**

The authors propose a token-based network expansion and modulation method to reduce negative transfer in multi-task problems.
When the task-specific gradients clash, modulating existing tokens is employed; when the task-specific gradients conflict, new task-specific tokens are added. The authors attempt to establish theoretical results to show that their proposed method can reduce training loss. Experiments are conducted on NYUD-v2, PASCAL, and taxonomy to evaluate their method.

**Strengths:**

- the paper analyzes the range space and null space of tokens to establish their method for reducing the negative transfer.
- Analyzing the conflicts between tasks by examining the token’s range space and null space is interesting.

**Weaknesses:**

- many important baselines are missing, e.g., InvPT, MQTransformer, TaskPrompt, ForkMerge, MTMamba (https://arxiv.org/pdf/2407.02228)
- The results reported in table 1 are much lower than current sota methods (refer to the results reported in Table 1 of MTMamba, https://arxiv.org/pdf/2407.02228), I copied some results here:

(Semseg and Boundary: larger is better; Depth and Normal: smaller is better)
method Semseg Depth Normal Boundary

InvPT 53.56 0.5183 19.04 78.10

MQTransformer 54.84 0.5325 19.67 78.20

MTMamba 55.82 0.5066 18.63 78.70

TM+TE   38.27 0.6370  21.64  57.90

we can see from the above results, the TM+TE  method proposed in this paper is much weaker than existing sota methods.

- the method is limited to transformer-based networks, cannot be extended to other architectures like CNN and mamba-based
- the proofs of theorem 1 and theorem 2 are problematic. For a mathematical proof, first, we need to make assumptions in the theorem, instead of making the assumption in the proof, e.g., L819, in cases; eq (5) $\approx$, approximation in eq (6).
moreover, the derivative in L837 is not rigorous.

**Questions:**

- does the proposed algorithm converge?

---

### Official Review · Reviewer_ZcwA · 2024-11-03

**Soundness:** 3
**Presentation:** 3
**Contribution:** 3
**Rating:** 5
**Confidence:** 2

**Summary:**

This paper proposes a novel approach called Dynamic Token Modulation and Expansion (DTME-MTL) for multi-task learning with transformer architectures. The key idea is to dynamically expand the network by manipulating tokens to mitigate negative transfer between tasks. The authors provide a theoretical analysis of gradient conflicts in the token space and propose two techniques - token modulation and token expansion - to address conflicts in the range space and null space respectively.

**Strengths:**

- The paper provides a solid theoretical analysis of the effectiveness of the proposed approach. The authors carefully define the token space using SVD and provide mathematical justification for categorizing gradient conflicts into range space and null space conflicts. This theoretical grounding helps explain why the proposed token modulation and expansion techniques should work.
- The method is general and can be applied to existing transformer-based multi-task architectures in an off-the-shelf manner. This increases the potential impact and applicability of the work.
- The proposed techniques of token modulation and expansion are novel and intuitively motivated. The idea of dynamically expanding the network via tokens rather than entire layers is interesting.

**Weaknesses:**

- While the authors claim their method is parameter-efficient, there is a lack of concrete efficiency comparisons with baselines. The paper would be strengthened by including quantitative comparisons of parameter counts and computational costs against other multi-task optimization approaches.
- The paper mentions that the number of layers for expansion is an important hyperparameter (Section 4.3), but there is no discussion or ablation study on how this hyperparameter affects performance. More analysis on the sensitivity to this choice would improve the paper.
- The experimental evaluation, while showing improvements, is somewhat limited. More extensive comparisons on additional datasets and task combinations would help establish the generality of the approach.

**Questions:**

- How sensitive is the method to the choice of the r parameter used to divide the range and null spaces? Is there a principled way to set this?
- Have the authors explored applying their method to other architectures beyond transformers? Could the core ideas be extended to CNNs for example?
- How does the computational overhead of computing SVD and projecting gradients at each layer impact training time compared to baselines?

---

### Official Review · Reviewer_eXAe · 2024-11-03

**Soundness:** 3
**Presentation:** 2
**Contribution:** 2
**Rating:** 5
**Confidence:** 4

**Summary:**

The paper tackles the problem of multi-task learning (MTL), more specifically in the context of designing MTL architectures. Such architectures usually rely on task-specific vs shared features. The main hypothesis of the paper is that defining such architectures in a fixed manner is suboptimal, and they should instead adapt dynamically to the tasks at hand.

To do this, the proposed method estimate **gradient conflicts** between tasks, but in a more fine-grained manner than previous work. First, the proposed method performs a SVD on the token space, sorting the resulting space by eigenvalues, which are then used to divide it into two parts: the range space (high eigenvalues) and the null space (low eigenvalues). Then, the task gradients are projected on each of these two subspaces, resulting into two different types of gradient conflicts (in the range space or in the sound space).

Based on the type of conflict, the architecture can be modified in two ways:
  * token modulation (conflict in the range space): Addition of an extra linear transformation of the shared tokens
  * token expansion (conflict in the null space): Addition of extra task-specific tokens

The proposed method is evaluated on NYU-D and PASCAL (3 to 5 tasks) and on a subset of Taskonomy (11 tasks)

**Strengths:**

* The method description and accompanying figures are clear
* Comparison on a varied number of tasks and with different degrees of task interference
* Interesting ablations such as when (during training) to start token expansion

**Weaknesses:**

## Literature review
My main concern with the paper is the literature review and justification of the method. In particular, there is a lot of work on dynamic architectures for MTL not mentioned in the paper, and, based on the introduction, there’s also little motivation for introducing the separate range and null space to perform the conflicting gradient analysis.

  * Dynamic architectures for MTL in vision is not a new idea in particular for CNN architectures, and many of such methods could be readily adapted to ViT. Examples of such literature:
    * "Efficiently Identifying Task Groupings for Multi-Task Learning" and “Which tasks should be learned together in multi-task learning?” -> design separate encoders for (learned) subgroups of tasks.
    * “Stochastic filter groups for multi-task cnns: Learning specialist and generalist convolution kernels”
    * “Learning to branch for multi-task learning” -> Neural architecture search for MTL
    *  “Latent multi-task architecture learning”
    * “Learning multi-level task groups in multi-task learning”

  * Optimisation vs architecture-based MTL: Many of the baselines in experiments are optimisation-based ones (e.g. GradNorm, CAGrad etc) which mainly deals with reweighing the various task losses. However, these methods are rather orthogonal with the proposed method, since we could combine any architectural changes with loss scaling.

  * **CNN baselines:** While the proposed method is compared against MTL-CNN models, it is not clear to me whether the experimental setup are equivalent. In particular, the proposed method starts from a ViT backbone pretrained on IamgeNet-22k. In contrast, to the best of my knowledge, MTI-Net models use pretrained ImageNet-1k weights. Even if the authors do not retrain these models from scratch, it would be interesting and fair to report a summary of experimental/pretraining difference between the different baselines to the reader.


## Experiments

  * **Model efficiency** should be discussed since we are dealing with dynamic architecture. The only efficiency metric is number of parameters (line 514) is model parameters, which seems a bit misleading since the method is also adding tokens. Because of this, it is really hard to assess how practical the method actually is. In particular it would be interesting to discuss:
    * The training time cost of performing SVD + gradient conflict estimation
    * The cost of the final model, with the additional tokens and modulation layers
  This would be particularly interesting for **Table 4**, as for instance we see that DTE-MTL only brings little improvement to `TaskPrompter`, and it would be interesting to see how this contrast with any potential overhead.

  * Impact of **hyperparameters**. As noted in “*In Defense of the Unitary Scalarization for Deep Multi-Task Learning*” and *“Do Current Multi-Task Optimization Methods in Deep Learning Even Help?”* (both published in NeurIPS 2022), the choice of learning rate is highly impactful on MTL methods, sometimes even leading to simple scalarization outperforming more advanced MTL optimisation. Yet in the current paper, the results are only reported for a single learning rate (line 409) and there is no mention of a learning rate sweep

**Questions:**

* In line 150: *"Existing methods that use pre-defined architectures for MTL have limitations in reducing negative transfer since they cannot preemptively prevent the occurrence of conflicting gradient"* -> This should be substantiated. The goal of having task-specific parameter (i.e. non-shared) is precisely to have parameters that can not be affected by conflicting gradients.

---

### Official Review · Reviewer_YC8T · 2024-11-05

**Soundness:** 3
**Presentation:** 3
**Contribution:** 3
**Rating:** 6
**Confidence:** 4

**Summary:**

This paper proposes a novel multi-task expansion algorithm for transformer architectures. The intuition is to dynamically expand the token representations of the transformer based on gradient conflict. Further, gradient conflict is separated into two distinct regions using singular-value decomposition of the shared representations: conflict which occurs in the null-space of the tokens and conflict which occurs in the range-space (i.e. where singular-values are non-zero).

The authors propose two different expansion techniques to address each type of conflict distinctly. When conflict occurs in the range-space of the token representations, the authors propose to add task-specific, learnable affine transformations to the token representations to prevent gradient conflict from slowing learning in the key range-space of the representation. Alternatively, when conflict occurs in the null-space of the token representations, the authors propose to add task-specific components to the token (i.e. instead of linearly transforming the representation, a new dimension is added to resolve conflict).

The authors demonstrate the efficacy of their method across 3 challenging multi-task settings, demonstrating it’s superiority compared to other multi-task network expansion algorithms, as well as multi-task optimization methods. They also present theoretical analysis of their method, demonstrating it’s ability to lower the multi-task loss, and empirical analysis of various components of their method.

**Strengths:**

The intuition to separate treatment of gradient conflict based on whether the conflict occurs in the subspace spanned by top eigenvectors vs. the subspace spanned by eigenvectors that are near zero is a neat idea, which (to my knowledge) has not explicitly been done before and is something that is worth considering in future work.

Additionally, the concept of focusing on the token representations in a transformer is novel, but certainly very parameter-efficient while also being effective, although it is worth noting that the modulation mechanism bears some similarity to RotoGrad [2], which applies learned task-specific rotations (linear transformation) to task representations.

The method appears to get strong results compared to an extremely extensive list of methods, including a number of multi-task optimizers and multi-task network algorithms. And the method is compared over 3 fairly common and difficult multi-task settings, so the results here are promising.

Finally, the empirical results contain a lot of analysis of various components of the proposed method which is useful to study, although they do not often have a clear takeaway (for example, the timing of network expansion and analysis of gradient conflict do not show particularly clear trends).

[2] Javaloy & Valera, 2022; ROTOGRAD: GRADIENT HOMOGENIZATION IN MULTITASK LEARNING

**Weaknesses:**

To me the greatest weakness of the work is a lack of analysis on ablations, particularly around the two separate mechanisms for dealing with gradient conflict. In particular, both mechanisms (modulation and expansion) add new, task-specific learnable parameters to the token representations where gradient conflict is highest. However, while the intuitions provided make sense, I do not see a principled reason why modulation could not be used when gradient conflict exists in the null-space, or why token expansion could not be used when conflict exists in the range-space. I’m not sure that the analysis in the theorems / appendix makes this clear either, e.g. if we assumed in the proof of theorem 1 that the token was spanned by the null-space, wouldn’t the modulation still lead to lower multi-task loss due to the task-specific parameters? Perhaps I am missing something here. Also, while there is an ablation study when using only token expansion vs. only modulation, I believe (please correct me if I am wrong) that token expansion is still only applied to null-space conflict and modulation is still only applied to range-space conflict, so this doesn’t address my concern.

This concern is compounded by the fact that applying modulation and token expansion to random or lowest-conflict layers still seems to improve over the multi-task baseline, suggesting that a key benefit may be the addition of more task-specific space to the model at the token-level, as opposed to the separation of range-space and null-space conflict and the corresponding mechanisms.

The lack of reporting on the number of random seeds or variation across those seeds in performance is also fairly problematic. These results would be much more convincing if we could see how significant the improvement is over a set of random seeds, especially as other works have shown that significance is a key detail to report in multi-task methods [1].

Finally, the method seems to rely on a shared input for all tasks in order to have a shared token representation. While this certainly works for a number of multi-task settings, it does exclude a larger number of MTL settings where distinct tasks have distinct inputs, and thus there is no single shared representation for each task input during training. However, I don’t see this limitation discussed or acknowledged anywhere.

[1] Kurin et al., 2022; In Defense of the Unitary Scalarization for Deep Multi-Task Learning

**Questions:**

See weaknesses.

---

### Note · Authors · 2024-11-14

**Comment:**

Thank you to all reviewers for their sincere feedback and efforts. Unfortunately, we have decided to withdraw our paper.

**Withdrawal Confirmation:**

I have read and agree with the venue's withdrawal policy on behalf of myself and my co-authors.